# Exploring the Role of Artificial Intelligence in Facilitating Assessment of Writing Performance in Second Language  Learning

Zilu Jiang [1,*], Zexin Xu [2], Zilong Pan [3], Jingwen He [4] and Kui Xie [4]

1 Center for Social Organization of Schools, School of Education, Johns Hopkins University, Baltimore, MD 21218, USA
2 Department of Computer Science and Engineering, College of Engineering, The Ohio State University, Columbus, OH 43210, USA; amoyxu44@gmail.com
3 Teaching, Learning, and Technology Program, College of Education, Lehigh University, Bethlehem, PA 18015, USA; zip322@lehigh.edu
4 Department of Counseling, Educational Psychology and Special Education, College of Education, Michigan State University, East Lansing, MI 48824, USA; hejw@msu.edu (J.H.); xiekui@msu.edu (K.X.)
* Correspondence: zjiang72@jh.edu

**Abstract:** This study examined the robustness and efficiency of four large language models (LLMs), GPT-4, GPT-3.5, iFLYTEK and Baidu Cloud, in assessing the writing accuracy of the Chinese language. Writing samples were collected from students in an online high school Chinese language learning program in the US. The official APIs of the LLMs were utilized to conduct analyses at both the T-unit and sentence levels. Performance metrics were employed to evaluate the LLMs' performance. The LLM results were compared to human rating results. Content analysis was conducted to categorize error types and highlight the discrepancies between human and LLM ratings. Additionally, the efficiency of each model was evaluated. The results indicate that GPT models and iFLYTEK achieved similar accuracy scores, with GPT-4 excelling in precision. These findings provide insights into the potential of LLMs in supporting the assessment of writing accuracy for language learners.

**Keywords:** large language models (LLMs); artificial intelligence (AI); writing accuracy; T-unit; sentence level analysis; language learning

## 1. Introduction

Writing accuracy refers to the ability to produce written content that is error-free (Wolfe-Quintero et al. 1998). It serves as an important indicator of a learner's ability to effectively utilize the target language. It can be evaluated through a range of measures, encompassing error-free units, error counts, specific error types and assessments based on the severity of errors (Peng et al. 2020).

T-unit-level and sentence-level analysis are two common approaches for identifying error-free units for assessing writing accuracy (Evans et al. 2014). T-units stand for "terminable units". A T-unit represents the smallest identifiable unit into which sentences can be broken down (Hunt 1965). Based on Hunt's (1965) definition of T-units in English, Jiang (2013) defined T-units in Chinese as "A single main clause that contains one independent predicate plus whatever other subordinate clauses or non-clauses are attached to, or embedded within, that one main clause" (p. 5). When a compound sentence has two independent clauses that are either connected by coordinating conjunctions like 'and' or separated by a comma, it is considered as having two T-units (Jiang 2013). Sentences are marked by punctuation that writers use to express a complete thought, such as a period ".".

Nevertheless, evaluating a large number of writing excerpts produced by language learners requires a significant amount of time and may result in unintentional errors. Fortunately, current advancements in artificial intelligence and natural language processing

have unlocked the potential for developing writing assessments based on more powerful large language models (LLM) (Brown et al. 2020). LLM-incorporated tools provide the possibility to enhance the assessment of writing accuracy in second language instruction. The exceptional text-processing capabilities of LLMs can allow researchers or instructors to evaluate a large volume of writing excerpts more efficiently if they can provide accurate predictions. However, the robustness and efficiency of these AI tools in facilitating second language writing assessments remain unknown.

Therefore, the current study intends to explore the affordance of four LLMs, which are GPT-4, GPT-3.5, iFLYTEK Open Platform and Baidu AI Cloud, as they represent advanced and extensively utilized LLMs within the field. This study investigated their effectiveness in supporting the assessment of writing accuracy at the T-unit and sentence levels, aiming to generate practical implications to inform future uses of LLMs in writing assessments.

Although we acknowledge that writing quality extends beyond these aspects to include coherence, organization, cultural appropriateness, complexity and fluency, our study takes a focused approach to assess accuracy. The accuracy of word choice and grammar structure serves as the foundation of language acquisition. Language learners often begin by mastering basic vocabulary and sentence structures, making these elements crucial for effective communication and comprehension. They can also be quantitatively measured and compared across different language models and human assessments. This lends itself well to an evaluation of LLMs' capabilities and their potential to assist in language learning and writing assessments.

Therefore, this research seeks to provide insights into the following research questions:

1. How robust are LLMs (GPT-4, GPT-3.5, iFLYTEK Open Platform, and Baidu AI Cloud) in assessing writing accuracy in general at the T-unit level and sentence level?
2. How efficient are LLMs (GPT-4, GPT-3.5, iFLYTEK Open Platform, and Baidu AI Cloud) in assessing writing accuracy in general at the T-unit level and sentence level?
3. What types of discrepancies exist between human rating results and AI rating results?

## 2. Literature Review

Assessments are an essential and integral component of teaching and learning practices, as they allow educators to determine and understand the level of progress that students, whether as individuals or in groups, have achieved in their learning at a specific point in time (Zamen 2020).

The assessment of accuracy in the Chinese language typically encompasses the evaluation of character errors, lexical errors, and syntactic errors. These errors can be analyzed individually, focusing on aspects such as lexical accuracy and syntactic accuracy. Alternatively, a more holistic approach involves considering overall accuracy, which accounts for all types of errors. This approach has been widely used in previous literature (Liao 2020). For instance, in a study conducted by Jiang (2013), a broader perspective was adopted to analyze accuracy. The number of error-free T-units in each written sample was counted. An error-free T-unit was considered not only grammatically correct but also semantically appropriate within its context. Errors in characters were also counted, as they often led to confusion and grammatical issues. The researcher further recommended taking contextual factors into consideration when assessing the accuracy of T-units. Some T-units may appear accurate when considered in isolation but may become inappropriate when placed back into broader discourse. This approach provides a more comprehensive understanding of accuracy, incorporating both linguistic correctness and contextual suitability, thus offering a more nuanced assessment of language proficiency.

In Liao's (2020) research, a focused approach was taken toward the examination of lexicon and syntax. Errors in character writing were treated as a distinct skill. Liao's analysis centered on errors in content words (such as nouns, verbs, adjectives, adverbs, prepositions, conjunctions, pronouns and auxiliaries) and function words (including prepositions, conjunctions, pronouns and auxiliaries). Two main types of errors were considered for these lexical usages: word form errors and word choice errors. Word form errors were

characterized by issues related to the formation or structure of words, which encompassed instances like incorrect character order in forming a word or missing characters within a word. On the other hand, word choice errors pertained to the inappropriate usage of words within a given context.

Furthermore, the study also explored syntactic errors in the Chinese language. These errors are frequently observed in areas such as word order, structural usages and the presence or absence of constituents. Word order specifically concerns the sequence of words within a sentence, whereas structural usages encompass a wide array of aspects related to sentence formation, such as the proper use of structures, measure words, determiners, etc. Misusing these elements can lead to sentences that are grammatically incorrect. Furthermore, missing or redundant constituents can disrupt syntax, and forgetting essential components or including unnecessary ones may alter the sentences' intended meaning. The choice between a focused approach and a holistic approach depends on the specific research objectives and the depth of understanding required. Researchers can consider how different error analysis approaches complement one another and how they can be strategically integrated to provide a more comprehensive assessment of language proficiency.

Although researchers can manually go through all language samples to evaluate accuracy, human grading and assessments may be hindered by the possibility of subjectivity and a lack of consistency. More specifically, different graders may interpret the same work differently or have varying levels of expertise or familiarity with the subject matter, leading to differences in their interpretation and evaluation of student performance, which can result in discrepancies in the assigned grades (Montero et al. 2006). Human graders may be influenced by factors such as bias or fatigue, which can further impact the accuracy and consistency of the assessments. Additionally, another limitation is the time and resources required for human grading, which can be a significant burden for large-scale assessments (Aldriye et al. 2019). These variabilities can create a lack of uniformity and reliability in the grading process, potentially impacting the fairness of evaluating students' academic outcomes (Way et al. 2019). Therefore, in response to the limitations associated with human grading, educational institutions and instructors are actively exploring the integration of AI tools to accommodate the subjectivity, consistency and efficiency of grading large-scale assessments.

The emergence of AI and LLM-incorporated grading techniques may provide an opportunity to address issues of subjectivity, inconsistency and inefficiency that could arise with human grading. Through the utilization of predefined criteria and algorithms, these systems can deliver evaluations that are consistent and objective. This approach mitigates the risk of bias and discrepancies between graders, thereby fostering fairness and guaranteeing that students are assessed according to standardized measures. For example, Abd El-Haleem et al. (2022) applied an AI-incorporated performance assessment technique to support the automatic grading process. It was designed to automatically detect a student's performance during the experiment run time. The findings showed that the proposed technique outperforms previous assessment techniques in terms of both accuracy and efficiency. In another study, Kortemeyer (2023) utilized GPT-4 to grade written physics problem solutions. The solutions were scored on a rubric that included criteria such as correctness of approach and symbolic derivations. The rubric scores were combined to obtain a total score, with a stronger weight on the final numerical result. The outcomes showed that GPT-4 is capable of providing objective feedback that can be helpful in formative assessment scenarios and can be used for an initial round of grading that sorts and flags solution approaches in summative scenarios. The study also suggested that future research should involve authentic student work from exams in large-enrollment physics courses and compare the results from AI grading to those from a traditional grader pool to establish inter-rater reliability.

Another advantage of using AI for grading is the potential for increased efficiency and time savings (Abd El-Haleem et al. 2022). Automated grading systems can process a large volume of assignments or assessments in a fraction of the time that manual grading

would take (Aldriye et al. 2019). This is particularly beneficial in contexts in which timely feedback is crucial or when dealing with a large number of students. Multimodal AI techniques can be used for automatic short answer grading and student assessments, which could potentially be applied to written assignments. There has been an emergence of automated essay evaluation systems that utilize machine learning techniques to grade essays and written assignments. These systems apply different algorithms to analyze a corpus of textual entities and classify them into discrete categories that correspond to possible grades. The application of AI in grading offers the potential to address some of the inconsistencies and inefficiencies associated with human grading (Ramalingam et al. 2018).

In light of advancements in AI and LLM techniques, researchers are actively engaged in the enhancement of automatic grading systems to evaluate essays and written assignments, aiming to attain objectivity, consistency and efficiency in the assessment process. For example, Alqahtani and Alsaif (2019) examined the application of Automated Essay Scoring (AES) systems in the automatic evaluation of learners' essays. The AES systems analyzed the essays based on free textual essay analysis, irrespective of the presence of predefined model essays. The evaluation criteria encompassed surface-based and text processing elements, such as spelling, punctuation, essay structure, coherence and style. The findings revealed that the system achieved a reasonably accurate evaluation, with approximately 73% of the overall essay scores being correctly assessed. In another study, Hoblos (2020) found that latent semantic analysis (LSA), a machine learning technique, can be used to assess the conceptual content of essay-type answers, and the results showed a high correlation with human grading. This study suggested that LSA can be used as a tool for automated essay grading with high accuracy. However, as the author indicated, further research may be needed to fully understand the potential benefits of using LSA in this context.

Researchers also pointed out the challenges associated with AI grading. One major concern is the ability of AI systems to accurately assess complex or subjective aspects of writing, such as creativity, originality, or nuanced arguments. Zhang et al. (2022) proposed a novel creativity assessment model based on the pre-training of deep bidirectional transformers (BERT) that imports a token-level metaphor identification method to extract metaphors as the indicators for creativity scoring. The experimental results showed that the model can accurately assess the creativity of different texts with precise metaphor identification. This paper offered a novel perspective on the evaluation of automatic writing assessments, highlighting the significance of incorporating metaphorical expressions in the scoring of creativity. The proposed model presented a valuable opportunity to enhance the precision of automatic writing assessments and contribute to the advancement of computational creativity and its associated challenges.

In summary, prior empirical studies employing AI techniques have demonstrated promising results. Given the ongoing rapid progress of LLMs, it is expected that the utilization of LLMs for assessing writing assignments will become increasingly reliable and efficient, thus providing educators with a valuable tool for evaluation. This research aims to explore the potential benefits and limitations of LLMs in light of their rapid advancement. Specifically, our goal is to conduct a comprehensive evaluation of several prominent LLMs by assessing their performance in grading writing assignments produced by Chinese language learners. Through this assessment, we aim to gain deeper insights into the capabilities and limitations of these LLMs, thereby making a valuable contribution to the existing body of research on writing assessments using AI models.

## 3. Methodology

### 3.1. Context

We collected a total of 2197 T-units and 1410 sentences from 123 weekly writing excerpts provided by 41 Chinese learners participating in an online high school language program delivered by a Midwestern university in the US. Data were collected from students' asynchronous writing assignments throughout three weeks. Each week, students

were asked to complete a writing exercise using the newly learned language expressions. The language competency of the students ranged from the beginner level to the AP level. Students' writing excerpts were transformed into a spreadsheet for T-units and sentences organization and analysis.

The T-unit extraction process often relies on the subjective decisions of the researchers, by which researchers carefully segment sentences into small units. Two researchers worked together to extract T-units, ensuring that the T-units extracted were terminable and meaningful units. Discrepancies in the extraction process were resolved through discussions and agreement. Sentence identification was based on the punctuation marks used by students, such as periods or exclamation marks.

After the extraction process was completed, two native Chinese-speaking researchers manually graded the writing samples. We adopted a comprehensive approach to assess the overall accuracy of the writing excerpts. To guide our manual grading process, we combined the criteria put forth by Jiang (2013) and Liao (2020). Our evaluation of T-units was primarily based on three key categories: word, structure, and context.

For a T-unit to be considered correct, it must employ appropriate words and structures while remaining meaningful within its given context. In the word category, we identified three sub-types of errors. In addition to the word choice and word form errors mentioned by Liao (2020), we also incorporated errors in word writing, following Jiang (2013). In terms of structure, we retained the four aspects highlighted by Liao (2020): structure usage, word order, redundancy, and missing components.

Each researcher worked independently on grading the T-units. Correct units were marked as 1, whereas incorrect units were marked as 0. The inter-rater reliability score of the coding results reached 0.93.

### 3.2. Procedures

The investigation into the robustness and efficiency of four LLM tools, namely GPT-3.5, GPT-4, iFLYTEK, and Baidu Cloud, involved processing data through their respective official APIs (Application Programming Interfaces). APIs serve as the official entry point for researchers to access and utilize these LLMs. Researchers can send requests to the LLMs via the official API, receiving responses that contain the desired results.

GPT3.5 and GPT-4 are generative models, and as such, the returned result is a sentence. However, data preprocessing is required to classify the label of the sentence. On the other hand, iFLYTEK and Baidu Cloud are classification models, and their returned results are already classified as different types of language errors, requiring no further processing. The detailed process can be found at the following link: https://github.com/Asonjay/Chinese_Writing_Assessment, accessed on 22 May 2023.

### 3.2.1. GPT-3.5 and GPT-4

Applying for API access to GPT-4 necessitates signing up for a waitlist and awaiting approval from OpenAI. In the current study, it took approximately 27 days to receive approval. However, GPT-3.5 does not require any official permission, and OpenAI provides detailed API documentation (https://platform.openai.com/docs/introduction, accessed on 22 May 2023). The current study incorporated prompts that inquire whether the language unit contains any language errors. To leverage context information, the request included the previous interactions of inquiries when processing T-units. We compiled all inquiries into a batch format and transmitted them to OpenAI via the appropriate API. Throughout this process, rate limit restrictions and connection errors were encountered, leading to prolonged processing times and an increased number of attempts for inquiries. The returned response from GPT models required manual coding. The researchers assigned labels to each response based on keywords. Responses that captured language errors were label as 0; responses that indicated correctness were labeled as 1.

### 3.2.2. iFLYTEK and Baidu Cloud

No permission is required to use the iFLYTEK API (https://www.xfyun.cn/doc/asr/lfasr/API.html, accessed on 22 May 2023) and Baidu Cloud API (https://cloud.baidu.com/doc/NLP/s/tk6z52czc, accessed on 22 May 2023). Both platforms provide detailed documentation for researchers. Language units can be sent directly to these models, and the responses include possible language error types present in the language unit. Similarly, a response was labeled as 1 when no language error was returned, whereas a label of 0 indicated the presence of an identified error.

### 3.3. Data Analysis

To address the first research question concerning the robustness of different models, a $2 \times 2$ confusion matrix (Table 1) was utilized to provide measures of model accuracy, precision, recall, and F1 score. Throughout the human assessment process, each T-unit was assigned a label of 1 for correctness and 0 for incorrectness. These human assessment results were subsequently used as the ground truth and compared against the outcomes produced by each LLM tool. For example, if the LLM tool inaccurately categorized a correct T-unit as incorrect, it did not align with the ground truth. The models were evaluated using the formula for each measure based on the results of TP, FP, TN and FN, and a robust model is expected to have all scores that are closer to 1 (Table 2). Regarding the second research question about efficiency, it was evaluated through the cost and time for processing the analysis. A model that demands a shorter processing time at a lower cost would be considered more efficient.

**Table 1.** Confusion matrix.

| Confusion Matrix | | Ground Truth | |
|---|---|---|---|
| | | **Actual Positive** | **Actual Negative** |
| AI Results | Positive | True Positives (TP) | False Positives (FP) |
| | Negative | False Negatives (FN) | True Negatives (TN) |
| True Positives: | | The outcome in which the AI correctly predicts the positive class. If the writing result was marked as correct by researchers, the AI also predicted it as correct. | |
| False Positives: | | The outcome in which the AI incorrectly predicts the positive class. The AI determined the result was true when it was false. In our context, if the writing result was marked as incorrect by researchers, the AI still predicted it as correct. | |
| False Negatives: | | The outcome in which the AI incorrectly predicts the negative class. The AI determined the result was false when it was true. In our context, if the writing result was marked as correct by researchers, the AI predicted it as incorrect. | |
| True Negatives: | | The outcome in which the AI correctly predicts the negative class. If the writing result was marked as incorrect by researchers, the AI also predicted it as incorrect. | |

For the third research question, content analysis was performed to summarize the types of discrepancies between human assessment results and LLM assessment results. The analysis results can help us identify categories of the error types and compress a large number of errors into fewer thematic categories. Researchers employed both deductive and inductive coding procedures to analyze the discrepancies (Cho and Lee 2014). A deductive approach was employed to analyze the false positive (FP) results. We utilized the word, structure and context categories along with their respective subcategories to guide the initial coding process. In cases in which two coders did not reach an agreement, revisions were made to the coding. Conversely, an inductive approach was employed for the analysis of the false negative (FN) results, which involved the open coding of error types, generating coding categories and revising codes. When the LLM tools incorrectly categorized the correct results as incorrect, it was considered overcorrection. We categorized the correction suggestions provided by the LLM tools into four types: overcorrection

with correct suggestions, overcorrection with incorrect suggestions, correction of other languages and need context to understand (See Tables 5 and 7 for explanations of each category and examples).

**Table 2.** Evaluation metrics.

| Evaluation Metrics | | |
|---|---|---|
| **Measure** | **Formula** | **Focus** |
| Accuracy | (TP + TN)/(TP + FP + TN + FN) | The number of correct predictions over all predictions. Accuracy is a good measure if a balanced classifier is presented, as it is interested in all types of outputs equally. |
| Precision | TP/(TP + FP) | The degree to which the AI made correct positive predictions out of all AI positive results. It indicates how many of the positive predictions made are correct. This metric is especially useful when we want to minimize false positives. In our context of assessing accuracy, we want to see the correct results as much as possible. However, we do not want to misclassify true errors into correct ones. Missing an error has a large cost in our context. Therefore, we wish to aim to maximize precision. |
| Recall | TP/(TP + FN) | The degree to which the AI correctly identified the positive instances out of the total number of actual positives. It indicates how good the model is at picking the correct items. Recall is an important indicator when we want to minimize the chance of missing positive cases. |
| F1 | 2*Precision*Recall/(Precision + Recall) | The harmonic mean of precision and recall, working well when the classifying results are unbalanced. |

Two researchers underwent coding training first to reach a consensus on the coding categories and independently coded the units. Recalibration was performed to ensure agreement and consistency in coding. The iteration of the recalibration process allowed us to clarify definitions, refine codes and address discrepancies.

## 4. Results

### 4.1. Robustness of LLMs

4.1.1. Human Assessment Results

Researchers identified 1656 correct and 541 incorrect T-units (correct rate = 75.38%) and 949 correct and 461 incorrect sentences (correct rate = 67.30%). The average number of correct T-units per writing excerpt was 13.46, and the average number of correct sentences per writing excerpt was 7.72. The findings indicate that, on average, the writing excerpts demonstrated a relatively high level of correctness in terms of T-units, along with a slightly lower but still above-average correctness rate for sentences.

4.1.2. LLM Assessment Results

The confusion matrix and evaluation metrics for T-unit results are presented in Figure 1 and Table 3. GPT-4, GPT-3.5 and iFLYTEK had the same accuracy score of 0.81. The accuracy score represents the overall correct prediction performance. These three models had higher matched result percentages than Baidu. iFLYTEK achieved the highest number that matched with human ratings, which was 81.3%, followed closely by GPT-3.5 (80.6%) and GPT-4 (80.1%).

The precision score is a measure of how accurately a system identifies true positives (TP) while minimizing false positives (FP). Within the scope of our study, which centers around evaluating writing accuracy, failing to identify errors represents a significant drawback. Models that fall short in detecting an adequate number of writing errors resulted in higher incidences of false positives. Therefore, minimizing the occurrence of false positives is crucial in order to achieve a high precision score.

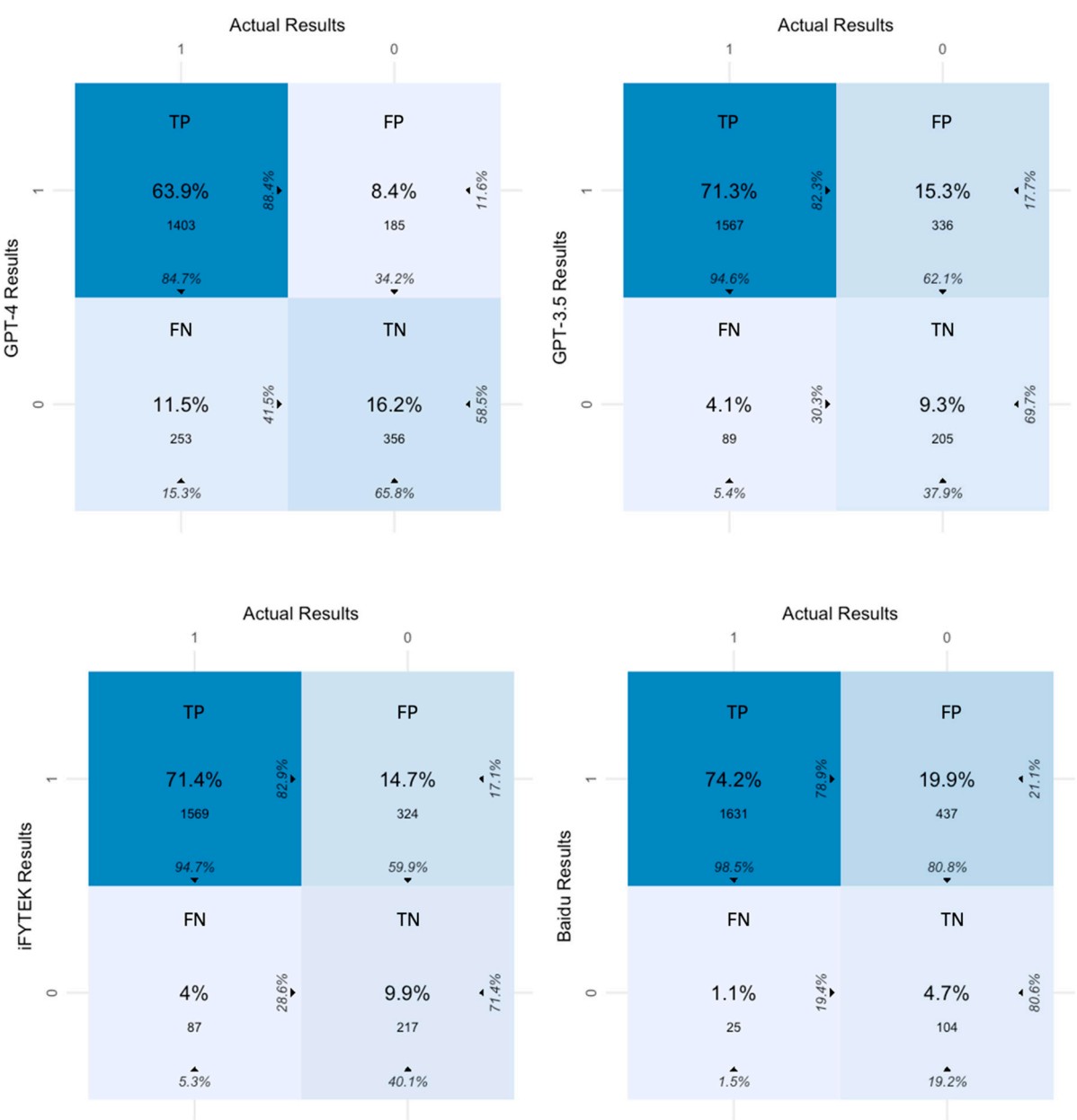

**Figure 1.** T-unit-level confusion matrix results.

**Table 3.** Evaluation metrics results.

|  | Matched Results (TP + TN) % | Unmatched Results (FP + FN) % | Accuracy | Precision | Recall | F1 |
|---|---|---|---|---|---|---|
| **T-units** |  |  |  |  |  |  |
| GPT-4 | 80.1 | 19.9 | 0.81 | 0.88 | 0.85 | 0.87 |
| GPT-3.5 | 80.6 | 19.4 | 0.81 | 0.82 | 0.95 | 0.88 |
| iFLYTEK | 81.3 | 18.7 | 0.81 | 0.83 | 0.95 | 0.88 |
| Baidu | 78.9 | 21 | 0.79 | 0.79 | 0.99 | 0.88 |
| **Sentences** |  |  |  |  |  |  |
| GPT-4 | 75.9 | 24.1 | 0.76 | 0.84 | 0.80 | 0.81 |
| GPT-3.5 | 71.3 | 28.7 | 0.71 | 0.71 | 0.96 | 0.82 |
| iFLYTEK | 77.3 | 22.7 | 0.77 | 0.78 | 0.93 | 0.85 |
| Baidu | 73.8 | 26.2 | 0.74 | 0.72 | 0.97 | 0.84 |

Among the evaluated models, GPT-4 demonstrated the highest performance with a score of 0.88, accompanied by the lowest false positive (FP) rate of 8.4%. GPT-3.5 and iFLYTEK exhibited similar scores of 0.82 and 0.83, respectively, with GPT-3.5 having an FP rate of 15.3% and iFLYTEK having 14.7%. Baidu had a relatively lower precision score compared to other models.

The recall score captures the ability of a model to accurately identify true positives, representing its sensitivity in detecting positive instances. The rate of FNs impacts the recall score. FNs are those that are actually positive but are mistakenly classified as negative by the model. Baidu achieved the highest recall score of 0.99. GPT-3.5 and iFLYTEK closely followed with recall scores of 0.95. On the other hand, GPT-4 achieved a relatively lower recall score of 0.85, suggesting that it may have missed more positive instances compared to the other models.

The F1 score combines the measures of precision and recall. The results indicate a relatively high level of balance between precision and recall for all models, ranging from 0.87 to 0.88.

In short, GPT-4 excelled in precision, and GPT-3.5 and iFLYTEK demonstrated comparable performance in terms of all measures. Although Baidu showed the highest recall score, it is important to consider its lower accuracy and precision scores.

The sentence-level analysis results are shown in Figure 2 and Table 3. Interestingly, the analysis revealed that using sentences as units for analysis is not advantageous. Compared to the T-units results, the overall performance scores were all lower across the four measures except the recall rate of GPT-3.5.

In terms of the accuracy score, GPT-4 and iFLYTEK achieved similar scores of 0.76 and 0.77, respectively. iFLYTEK was found to have the highest rate of matched results. GPT-3.5 had a lower accuracy score compared to other models. Regarding the precision score, similar to the patterns in the T-units analysis, GPT-4 achieved the highest precision than other models, followed by iFLYTEK. Although the precision score was lower than that in the T-unit level analysis, GPT-4 still remained above 0.8, which was even higher than the precision scores for the other three models in the T-units analysis. GPT-3.5 and Baidu demonstrated comparable scores. Additionally, the results revealed a consistent pattern in the recall scores, with GPT-4 achieving the lowest score and Baidu obtaining the highest score. The recall scores for GPT-3.5 and Baidu were found to be very close. Finally, iFLYTEK demonstrated a higher F1 score than other models. Overall, GPT-4 demonstrated consistent performance in achieving precision when using sentences in the analysis, and other models tended to identify more positive cases than GPT-4.

### 4.2. Efficiency of LLMs

The evaluation of processing time and costs is summarized in Table 4. In terms of the processing time, iFLYTEK exhibited the highest efficiency, completing all inquiries in less than half an hour. Baidu showed a similar speed in processing sentence inquiries, with a slightly longer time for processing T-units. On the other hand, GPT-4 exhibited the longest processing time, exceeding six hours to complete all inquiries. It is worth noting that GPT-4 was used with the paid ChatGPT Pro-service, which has a limit of 300 inquiries per hour.

Regarding the inquiry procedures, iFLYTEK and Baidu provided immediate results in response to the inquires. The results were downloaded from their platforms. The complexity of GPT models posed challenges during the study, as rate limit errors were encountered due to the model's capacity. Inquiries using GPT-3.5 encountered connection errors and rate limit issues, resulting in longer processing time due to disruptions and delays. In total, 15 queries were used to interact with language models to obtain the responses for processing all T-unit data. GPT-4 displayed the least efficient performance, characterized by extended processing durations attributed to rate limit problems and the intricacies of its language model. Eight queries were used to process all T-unit data.

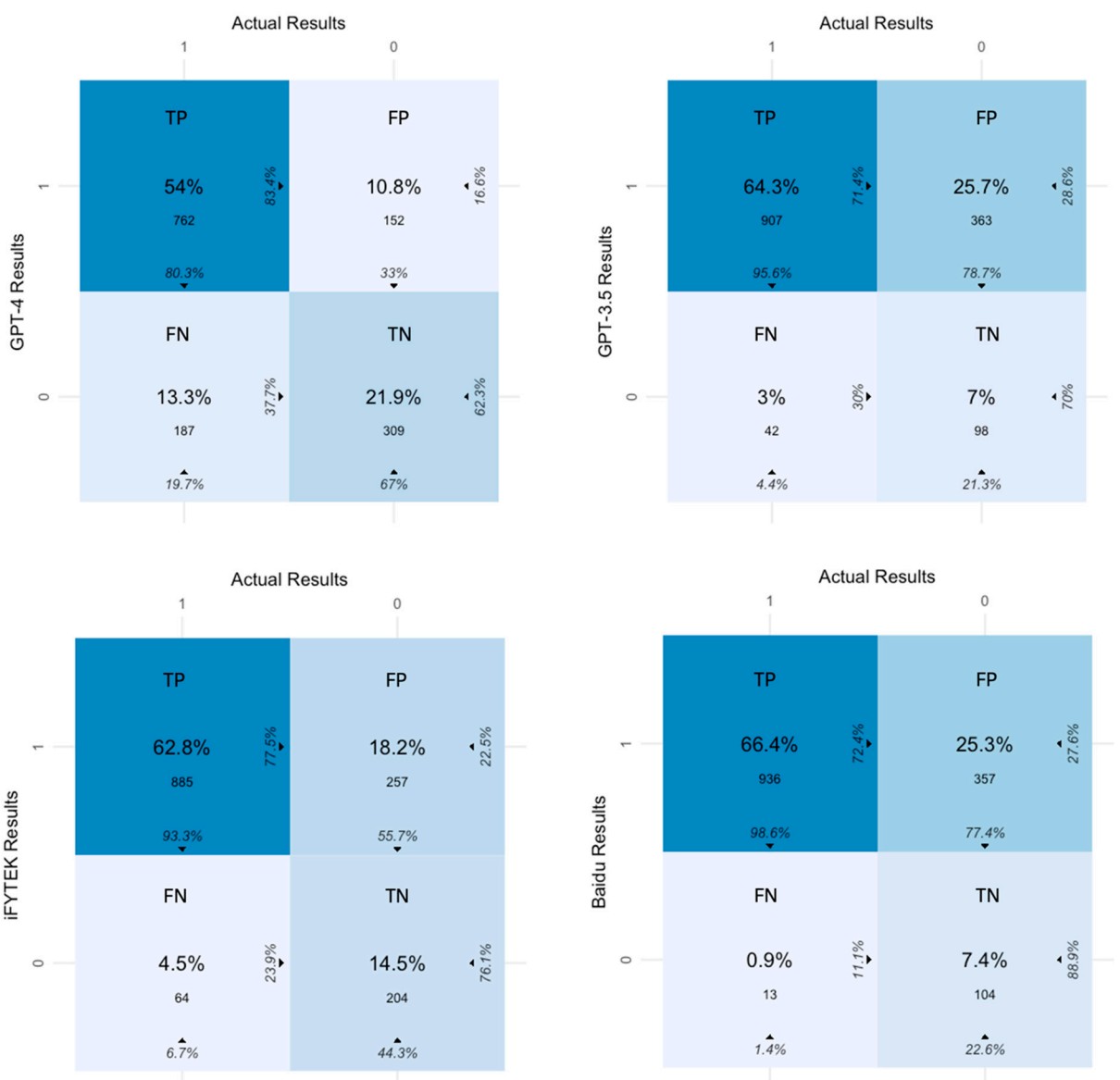

**Figure 2.** Sentence-level confusion matrix results.

**Table 4.** Comparison of cost and processing time for different models.

|  | GPT-4 | GPT-3.5 | iFLYTEK | Baidu |
|---|---|---|---|---|
| **T-unit** |  |  |  |  |
| Time | 6 h 31′39.6″ | 65′3″ | 27′14″ | 46′10.5″ |
| Query | 8 | 15 | 1 | 1 |
| Cost | USD 11.27 | USD 0.4 | 0 | 0 |
| **Sentence** |  |  |  |  |
| Time | 3 h 31′51″ | 27′23.4″ | 26′54″ | 29′54.4″ |
| Query | 2 | 1 | 0 | 0 |
| Cost | USD 5.78 | USD 0.17 | 0 | 0 |

In terms of the total cost, iFLYTEK and Baidu were more cost-friendly. No costs were accumulated because all queries that were sent aligned with the 1 million total word count restriction. However, exceeding the word count limit results in additional charges. GPT-3.5 was less expensive than GPT-4, with a total cost of USD 0.57. In contrast, the total cost for GPT-4 was around USD 17.

### 4.3. Discrepancies between Human and LLM Assessment Results

Baidu was excluded in the analysis, as it lacked information on error types. To gain a deeper understanding of the false positive (FP) and false negative (FN) results, qualitative coding was conducted. This qualitative coding aimed to examine the specific error types and patterns in the FP and FN cases. By analyzing these errors, we can understand the characteristics of the mistakes made by the models, identifying areas of improvement in model performance in the future. iFLYTEK and GPT-4 results were included in the final coding process. GPT-4 was chosen because it had the highest precision scores and comparable overall performance than GPT-3.5.

FP errors were categorized into word, structure and context (Tables 5 and 6). The number of FP errors was less in GPT-4. Out of the 185 FP results generated by GPT-4, the majority of the errors were attributed to grammar structure and word error types. Regarding the 324 FP results generated by iFLYTEK, most of the undetected errors were related to grammar structure. A similar trend in the results was revealed in the top numbers of the error types in two LLM results. The most undetected errors in grammar structure were primarily associated with missing components or redundancies. On the other hand, the highest number of errors in the word category were related to improper word choice. Both models showed error types that related to context, for which context was needed to be able to produce more accurate predictions, but the percentages were relatively lower.

For FN errors, four categories were identified (Tables 7 and 8), with both LLMs showing overcorrection. GPT-4 had a higher number of FNs than iFLYTEK. iFLYTEK error types were found to be related to overcorrection only. GPT-4 had few incorrect suggestions and could correct languages besides Chinese if the sentence that the second language learner wrote had other languages (e.g., English name). It also had a unique category for which it remained unsure and asked for context to judge accuracy. Despite having more FNs, GPT-4 focused on improving language and was conservative when there was insufficient context information. Interestingly, about 69% of the overcorrections made by GPT-4 were related to correct suggestions of language use, and it provided standard and appropriate language recommendations. Of overcorrections, 8% were incorrect, indicating the true presence of undetected positive cases, and this was the true mistake made by the model's predictions. It identified errors in the language units when they were actually correct.

**Table 5.** Error types for FP results.

| Category | Error Types for FP Results | Notes | Error Examples | Corrected Examples |
|---|---|---|---|---|
| Word | Word choice | A word choice error occurs when a word is used incorrectly or inappropriately in a unit, causing confusion or a change in meaning. | 我吃**的**以后觉得胃不太舒服 (My stomach does not feel well after I had the food.) | 我吃**了**以后觉得胃不太舒服。 (The use of 了 indicates the completion of the eating action.) |
| | Error in word writing | An error in word writing can be found when the word is written in an incorrect form. Usually, the wrongly written word is similar to the correct form of the word or has a similar sound. | 去看电影和吃**完**饭 (Go to watch a movie and have dinner.) | 去看电影和吃**晚**饭 (Although the pronunciations for 完 and 晚 are the same, the tones are different.) |

**Table 5.** *Cont.*

| Category | Error Types for FP Results | Notes | Error Examples | Corrected Examples |
|---|---|---|---|---|
| | Word form | A word form error can be found when characters are missing or the order of characters are incorrect. | 然后我**坐租车**到饭馆(Then, I took a taxi to the restaurant.) | 然后我坐**出**租车到饭馆 (The character **出** is missing.) |
| Structure | Structure usage | A structural usage error occurs when the structure of a sentence or phrase is incorrect or inappropriate. | 这个菜做**的**不难 (This dish is not difficult to make.) | 这个菜做**起来**不难 (Using "做起来" is the correct use of the direction complement.) |
| | Word order | A word order error refers to an error in the order of words or structure components. | 他**的做**饭很好吃 (His cooking is delicious.) | 他**做的**饭很好吃 ("做的饭"means the food he made.) |
| | Redundancy | A redundancy error occurs when unnecessary words duplicate the information expressed by other words or structure components in the unit. | 我在加拿大**里**住了十五年了 (I have been living in Canada for 15 years.) | 我在加拿大住了十五年了. (里 means in, but using this word is not necessary.) |
| | Missing components | Essential structural elements are missing. | 这是我书 (This is my book.) | 这是我**的**书 (的 is missing, which should be used to indicate possession.) |
| Context | | The unit is accurate on its own but is incorrect when taking the context into account. This type of error includes the error types that occur in the word and structure categories. | 我也非常喜欢吃**种**子 (I like eating seeds.) | 我也非常喜欢吃**粽**子 (I like eating glutinous rice.) |

**Table 6.** Qualitative analysis results of FP errors.

| | | GPT-4 | | | | iFLYTEK | | | |
|---|---|---|---|---|---|---|---|---|---|
| | | Total | % | *n* | % | Total | % | *n* | % |
| Context | Word | 21 | 11% | 16 | 9% | 32 | 10% | 26 | 8% |
| | Structure | | | 5 | 3% | | | 5 | 2% |
| Word | Word choice | 76 | 41% | 53 | 19% | 128 | 40% | 108 | 33% |
| | Error in word writing | | | 14 | 8% | | | 12 | 4% |
| | Word form | | | 9 | 5% | | | 8 | 2% |
| Structure | Structure usage | 88 | 48% | 19 | 10% | 164 | 50% | 40 | 12% |
| | Word order | | | 10 | 5% | | | 31 | 10% |
| | Redundancy | | | 26 | 14% | | | 42 | 13% |
| | Missing components | | | 33 | 18% | | | 51 | 16% |
| Total | | 185 | | | | 324 | | | |

**Table 7.** Error types for FN results.

| Category | Definition | Student's Response | GPT-4 Examples | iFLYTEK Examples |
|---|---|---|---|---|
| Overcorrection with correct suggestions | The original sentence is already acceptable in its current form. However, the suggestion provided may help to enhance the sentence by providing more clarity, conciseness or precision in its meaning. | 别的中国餐我也喜欢 (I also like other Chinese foods.) | 正确的表达方式应该：“我喜欢其他所有中国菜。”或者“除了这道菜，我喜欢吃其他所有中国菜。” (The correct expression should be, "I like all other Chinese dishes" or "I like all other Chinese dishes except this one.") | “餐”字使用错误，应使用“菜”字。(The word “餐meal” is used incorrectly; the word “菜dish” should be used.) |
| Overcorrection with incorrect suggestions | The original sentence is already acceptable in its current form. However, the suggestion provided is incorrect. | 你好李东艾 (Hello, Li Dongai.) | 这句话有一些语病。首先，在拼写上应该是“李东霖”而不是“李东艾”。其次，语序上略显不顺。正常的表达可以是：“李东霖，你好！”或者“你好，李东霖！” (First, in terms of spelling, it should be “李东霖” instead of “李东艾”. Second, the word order is slightly awkward. A more natural expression would be “李东霖，你好！” or “你好，李东霖！”.) | |
| | | 中国饭馆在哪儿 (Where is the Chinese restaurant?) | | “哪”字使用错误，应使用“那”字。The word “where” is used incorrectly; the word “there” should be used.) |
| Correction of other languages | The Chinese sentences are correct, but corrections are needed for sentences in other languages. | 我喜欢你的story (I like your story.) | 这句话中的”story”没有被翻译成中文，所以显得有些不协调。可以改成：“我喜欢你的故事。” (The word “story” in this sentence is not translated into Chinese, so it seems a bit incongruous. It could be changed to “I like your story.) | |
| Need context to understand | Due to insufficient contextual information, a judgement cannot be made. | 另一个没有 (The other one did not. The context for this unit is 一个有珍珠，另一个没有：one has the boba, the other does not.) | 句子没有给出完整的意思和信息。 (The sentence does not give the full meaning and information.) | |

**Table 8.** Qualitative analysis results of FN errors.

| Category | GPT-4 | | iFLYTEK | |
|---|---|---|---|---|
| Overcorrection with correct suggestions | 174 | 69% | 54 | 62% |
| Overcorrection with incorrect suggestions | 20 | 8% | 33 | 38% |
| Correction of other languages | 49 | 19% | | |
| Need context to understand | 10 | 4% | | |
| Total | 253 | | 87 | |

## 5. Discussion and Conclusions

The present study explores the effectiveness of GPT-4, GPT-3.5, iFLYTEK and Baidu in assessing writing accuracy. The analysis focused on metrics such as accuracy, precision, recall and F1 score. Each metric offers unique insights into different aspects of model performance. We need to consider multiple measures to evaluate the model comprehensively. Although GPT-4 had more FNs, which led to a lower recall score, it had the lowest number of FPs at the same time. iFLYTEK and GPT-3.5 had higher recall and a lower precision score than GPT-4. Baidu, on the other hand, had a very low rate of FNs, and its FP rate was the highest. The discrepancy between a high recall score and lower accuracy and precision scores suggests a potential trade-off between correctly identifying positive instances and the presence of false positives. This trade-off highlights the need for a balanced evaluation and the consideration of the study context.

Areas for improvement in future AI-facilitated writing assessment programs can target grammar structure component detection and redundancies, as well as word usages. For now, although AI models can offer valuable assistance in detecting errors and providing suggestions, human evaluations remain crucial for nuanced language analysis, context-specific interpretations and making systematic assessments of writing accuracy.

The findings reveal insight about distinctions between the approach of human raters and LLMs when assessing writing. Human raters had a higher tolerance for variations in writing quality and can gain a more nuanced understanding of the writing by taking into account the context and overall effectiveness of communication. On the other hand, models like GPT-4 relied on predefined language patterns and tended to make more conservative judgments. As a result, these models were more prone to identify correct samples as incorrect, particularly when the expression used varied from the standardized norm. Adherence to a strict language pattern contributed to a high rate of FNs. To improve the accuracy of automated writing assessments, future research could explore ways to incorporate a broader range of language patterns to enhance LLMs.

Furthermore, the accuracy of the generated suggestions from the generative AI proves to be valuable, as it serves as a reliable reference for the appropriate use of language. This benefits teachers when providing feedback on students' writing, improving both the efficiency of the assessment process and the quality of feedback offered. Generative AI can also contribute to personalized learning experiences for language learning. It can provide a tailored approach by specifically targeting the issues identified in a learner's writing. Exposure to alternative language usage can enrich learners' language repertoire and enhance awareness to error correction. However, it is important to note that, although generative AI can be highly useful, human checking is still necessary at this point in time.

This study's results contribute to the growing body of research on LLM-based writing assessments. The high precision scores achieved by GPT-4 highlight its potential as a valuable tool for accurately identifying writing errors. This study has certain limitations, including the specific dataset and the prompting techniques used. The prompts employed in this study did not explicitly instruct the bot to focus on the context. Improving prompting techniques could help mitigate context-related errors and enhance the overall accuracy of the assessment. Future research should expand the scope of the analysis to include a wider range of writing genres, and larger sample sizes. Continued research exploring ways

to improve the recall of AI models while maintaining high precision can contribute to the advancement of LLM-driven writing assessment tools, ultimately benefiting both language educators and learners.

**Author Contributions:** Conceptualization, Z.J., Z.X., Z.P. and J.H.; Methodology, Z.J. and Z.X.; Software, Z.X.; Validation, Z.J., Z.X. and J.H.; Formal Analysis, Z.J. and Z.X.; Investigation, Z.J., Z.X., J.H. and Z.P.; Writing—Original Draft Preparation, Z.J., Z.X., Z.P. and J.H.; Writing—Review and Editing, Z.J., Z.X., Z.P., J.H. and K.X.; Visualization, Z.J.; Supervision, K.X. All authors have read and agreed to the published version of the manuscript.

**Funding:** This research received no external funding.

**Institutional Review Board Statement:** Not applicable.

**Informed Consent Statement:** Not applicable.

**Data Availability Statement:** Not applicable.

**Acknowledgments:** We acknowledged the contribution of AI-powered tools in facilitating grammar checks during the proofreading process.

**Conflicts of Interest:** The authors declare no conflict of interest.

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
