# Peer review of "Exploring the Role of Artificial Intelligence in Facilitating Assessment of Writing Performance in Second Language Learning"

_languages, doi:10.3390/languages8040247_

Round 1

Reviewer 1 Report

Thank you for the opportunity to review this manuscript. The current manuscript addresses the capability of different Large Language Models to assess the quality of Chinese writing. The authors utilized a mixed methods approach to analyze the accuracy of the LLMs’ quality judgments and their efficiency.

The overall quality of the manuscript is good. There is some room for improvement so I have presented my comments below, and organized them into major and minor comments.

Major comments:

There is much more that could be added to the literature review to explain how grammatical accuracy is only one small aspect of evaluating writing quality and justify why this is the only facet examined in this study. Additionally, what literature can be included to explain the process of evaluating writing quality in Chinese? What features are we looking for? Syntax? Word-formation? Word choice? Extending the review would provide useful context for the reader.

Please give the exact number of learners and how many weekly writing excerpts were produced per learner.

The explanation of the rating process needs to be expanded greatly. What were the two graders looking for to consider a sentence correct? What kind of errors or features constituted an incorrect sentence/unit?

The Data Analysis section is missing a bit of contextualization. Before explaining that model performance was evaluated using a 2 x 2 confusion matrix, please explain what kind of analysis you planned to conduct, explain the types of comparisons being made, and how the comparisons relate to the research questions. What does “The manually rated results” refer to? Is this a writing assignment and its scores? Is this a set of T-units? This information is present, but it is currently not organized in a way that gives the reader the clearest picture of the analyses.

Please elaborate on the “Human Assessment Results”. How many writings were analyzed across how many learners? What was the average number of correct sentences per writing? For a learner? What was the average number of correct T-units per writing? What can we know about these errors?

More explanation of the way efficiency is defined and operationalized. There is a way to feed data into OpenAI or GPT APIs in a batch format and get results much more quickly than 6 hours, so I am not sure why there is such a discrepancy. What was the exact process of feeding data into GPT? If you typed sentences/T-Units individually and got individual results each time, this is a very inefficient decision, though it might reflect what a teacher does. This process needs to be made clearer with justification provided.

Many of the terms in the qualitative results need to be explained in the methods in more detail. Please explain terms such as “word error”, “structure error”, “context error”, “overcorrection”. If you believe these definitions are indeed part of the researchers’ induction and part of the results, then much more information needs to be given in the methods regarding the theme-generating process of the qualitative analysis.

In the discussion, you mention that there are areas for improvement for AI-facilitated writing assessment. Could you explain how this might be done? To my knowledge, LLMs are not supervised in ways that manipulation of certain types of parsing or processing can be modified unless the training data dictates that it should change. What kind of actual improvements do you suggest?

Minor comments:

In the introduction, since Jiang (2013) is primarily restating Hunt’s definition of T-unit, though for Chinese, please indicate as such. The current definition is presented as though Jiang created a wholly new definition for T-unit.

In Table 4, how is the cost calculated? If it is related to the overall cost compared to the time needed, please make this clear.

On page 10, Table 5 is referred to in the text as Table 7. Please reconcile this.

On page 2, change “The study investigated their effectiveness in supporting assessing writing accuracy at T-units and sentences levels,” to “The study investigated their effectiveness in supporting the assessment of written grammatical accuracy at the T-unit and sentence levels,”.

On page 2, change the word “Assessments” at the start to “Evaluation”, or perhaps just stick with calling it grading rather than layering terms. Assessment is a much bigger term, encompassing more than grading and “level of progress”.

On page 4, change “T-units extraction process” to “The T-unit extraction process”, and change “Sentences indentifications” to “Sentence identification”. In general, be careful not to use plural nouns in the first noun of a noun-noun combination.

What does it mean that researchers ensured “that the T-units extracted were simple and meaningful units.”? I am not sure that T-units are necessarily simple. What does this mean?

In Table 2, change “It is a harmonic mean…” to “The harmonic mean…”

On page 10, iFLYTEK and Baidu are mentioned to be “more cost-friendly”. What was the exact cost? If they are free to use, then I suggest just saying that they are free. The cost of GPT-3.5 is currently listed as xx. Please provide the cost.

On page 13, The sentence “Despite the presence of unmatched results between LLM models and human assessment , our findings revealed the distinction in the approach of human raters and LLMs when assessing writing.” Is hard to understand. What do you mean by unmatched results? If you revealed the distinction, why is this “despite” the unmatched results? I am afraid I do not understand what is being introduced here.

Reviewer 2 Report

1. This study examined the robustness and efficiency of four large language models (LLMs), GPT-4, GPT-3.5, iFLYTEK, and Baidu Cloud, in assessing Chinese language writing accuracy.

2. The theme is original and addresses a specific gap in the field.

3. The methodology is well formulated and allows addressing the objective.

-Student writing samples were collected.

-The official LLM APIs were used to perform analyzes at both the T-unit and sentence level.

-Performance metrics were used to assess the performance of the LLM.

-LLM results were compared to human rating results.

-A content analysis was performed to categorize the types of errors and highlight discrepancies between human and LLM ratings.

-the efficiency of each model was evaluated.

4. The conclusions are consistent with the evidence and arguments presented and address the main question posed

- GPT-4, as was to be expected, exceeds precision.

- LLMs can support the assessment of writing accuracy for language learners.

5. The references are scarce, but current and adequate.

In my opinion, it is a good research paper. Congratulations

Author Response

Thank you so much for all the comments and feedback!